# Behavioral and Psychological Symptoms in Dementia (BPSD) and the Use of Antipsychotics

**DOI:** 10.3390/ph14030246

**Published:** 2021-03-09

**Authors:** Valeria Calsolaro, Grazia Daniela Femminella, Sara Rogani, Salvatore Esposito, Riccardo Franchi, Chukwuma Okoye, Giuseppe Rengo, Fabio Monzani

**Affiliations:** 1Geriatrics Unit, Department of Clinical & Experimental Medicine, University of Pisa, 56126 Pisa, Italy; v.calsolaro@ao-pisa.toscana.it (V.C.); sara_rogani@hotmail.it (S.R.); riccardo.franchi@me.com (R.F.); chukwuma.okoye@phd.unipi.it (C.O.); 2Division of Geriatrics, Department of Translational Medical Sciences, Federico II University of Naples, 80138 Naples, Italy; graziadaniela.femminella@unina.it (G.D.F.); salvatorespositomed@gmail.com (S.E.); giuseppe.rengo@unina.it (G.R.); 3Istituti Clinici Scientifici Maugeri SpA Società Benefit, Via Bagni Vecchi, 1, 82037 Telese, Italy

**Keywords:** dementia, behavioral and psychological symptoms of dementia (BPSD), atypical antipsychotics, typical antipsychotics, frailty

## Abstract

Dementia affects about 47 million people worldwide, number expected to exponentially increase within 30 years. Alzheimer’s disease (AD) is the most common dementia type, accounting on its own for almost 70% of all dementia cases. Behavioral and psychological symptoms (BPSD) frequently occur during the disease progression; to treat agitation, aggressiveness, delusions and hallucinations, the use of antipsychotic drugs should be limited, due to their safety issues. In this literature review regarding the use of antipsychotics for treating BPSD in dementia, the advantages and limitation of antipsychotic drugs have been evaluated. The available medications for the management of behavioral and psychological symptoms are the antipsychotics, classed into typical and atypical, depending on their action on dopamine and serotonin receptors. First generation, or typical, antipsychotics exhibit lack of tolerability and display a broad range of side effects such as sedation, anticholinergic effects and extrapyramidal symptoms. Atypical, or second generation, antipsychotics bind more selectively to dopamine receptors and simultaneously block serotonin receptors, resulting in higher tolerability. High attention should be paid to the management of therapy interruption or switch between antipsychotics, to limit the possible rebound effect. Several switching strategies may be adopted, and clinicians should “tailor” therapies, accounting for patients’ symptoms, comorbidities, polytherapies and frailty.

## 1. Introduction

The term “dementia” refers to a clinical syndrome characterized by progressive cognitive decline that interferes with independence in everyday activities. According to the Diagnostic and Statistical Manual of Mental Disorders (DSM)-IV, dementia is defined as a decline in memory and impairment in at least one other cognitive domain, such as executive function (e.g., planning, attention and abstract reasoning), language, visuospatial skills, praxis or judgment. Symptoms of dementia are gradual, persistent and progressive. Alongside the symptoms that affect cognitive functions, there are alterations in personality and behavior, such as agitation, apathy, aggression, psychosis, hallucinations and delusions; their clinical presentation varies greatly among individuals and can cause considerable distress for patients and their caregivers [1]. Alzheimer’s disease (AD) is the most frequent cause of dementia [2]. Other common causes of dementia include vascular dementia, dementia with Lewy bodies (LBD), frontotemporal dementia (FTD) and Parkinson’s disease dementia (PDD). There are also rare forms of degenerative brain diseases that can cause dementia, like progressive supranuclear palsy (PSP), cortical basal degeneration (CBD), multisystemic atrophy (MSA), Huntington’s disease (HD) and Creutzfeldt-Jakob disease (CJD) [3]. Mixed Vascular–Alzheimer Dementia (MVAD) consists in the simultaneous presence of Alzheimer’s disease and cerebrovascular disease (CVD); it is quite common in older patients, with a prevalence of around 22% [4]. Eventually, almost 10–15% of patients with cognitive impairment may have a reversible cause, if treated in time and correctly, like endocrine diseases, normal pressure hydrocephalus, depression and drug-induced dementia. Dementia is an increasingly common phenomenon due to the aging population, and a major global public health challenge; its prevalence is expected to exponentially increase within the next 30 years [5]. Nevertheless, a recent meta-analysis of age-specific incidence of dementia and AD found that incidence rates of dementia have declined over the last four decades, but AD incidence rate actually increased for younger age groups in non-Western countries [6]. Behavioral and psychological symptoms of dementia (BPSD) represent a heterogeneous group of affective, psychotic and behavioral symptoms that occur in the majority of patients with dementia, causing great suffering and increasing the caregivers’ burden [7]. In this review, we evaluate the use of antipsychotic drugs in dementia. The search was conducted in PubMed, with the keywords “Dementia, Alzheimer’s Disease, antipsychotic and Dementia/Alzheimer’s disease, BPSD, behavioral and psychological symptoms dementia, risperidone, quetiapine, haloperidol, aripiprazole, olanzapine, antipsychotics tapering, antipsychotics switch, typical/atypical antipsychotics”; meta-analysis, reviews and original works and abstracts in English have been selected and reviewed, published within the last 10 years. 

## 2. Behavioral and Psychological Symptoms of Dementia (BPSD)

BPSD are among the earliest signs and symptoms of neurocognitive disorders and cognitive decline and, although they fluctuate, their severity exponentially increases over the course of the disease. Neuropsychiatric symptoms are associated with several negative outcomes, such as faster cognitive decline, functional impairment, reduced independence and inability to complete activities of daily living, with progression to more severe stages of dementia and increasing risk for secondary complications such as falls and fractures, causing higher hospitalization rates and eventually early institutionalization [8]. The etiopathogenesis of BPSD is complex as it is probably the result of the interaction of multiple factors, such as biological (brain changes, comorbidities, medications), psychological (personal life history, personality) and social factors (support network, living arrangements) [8]. In their review, Eissa et al. examined the hypothesis that chronic neuroinflammation may be associated with cognitive deficits, and found that central histamine (HA) plays a significant role in the regulation of neuroinflammatory processes of microglia functions in numerous neuropsychiatric diseases such as BPSD [9]. In a meta-analysis by Qing-Fei et al., apathy resulted as the most common neuropsychiatric symptom reported in the Neuropsychiatric Inventory (NPI), followed by depression, aggression, anxiety, sleep disturbances, irritability, change in appetite, motor problems, hallucinations, delusions, disinhibition and euphoria [10]. Psychiatric symptoms like depression, irritability, agitation in cognitively normal subjects may also be predictors of possible more rapid cognitive decline. In their study, Banks et al. assessed the relationship between behavioral symptoms and emergence of mild cognitive impairment or dementia in older adults, over a 4-year period. The results suggested that anxiety and depressed mood moderately increased the risk of developing dementia, primarily Alzheimer’s disease, representing precursors to future cognitive decline [11]. The relationship between depressive symptoms and cognitive decline appears to be complex; whether depression is a very early manifestation of Alzheimer’s disease or increases susceptibility to it remains to be determined. Nevertheless, a large longitudinal study of people aged 50 to over 90 years showed that depressive symptoms were associated with a slight acceleration in cognitive decline in people aged 60–80 years, but there was no support for the hypothesis that there might be a bidirectional connection between depression and AD [12]. Different BPSDs are often co-present and can be clustered into distinct domains, suggesting that they should be considered as groups of symptoms rather than lonely symptoms, with each group reflecting a different prevalence, timeline, biological and psychosocial correlates. During the last few decades, several studies have been conducted with the aim of identifying possible AD sub-syndromes defined by combinations of different neuropsychiatric symptoms. Most of these studies included only patients with AD, whereas others included patients with various dementia subtypes. In their study, Canevelli et al. identified three clusters of symptoms: 1—“psychotic” cluster (“delusions” and/or “hallucinations” items); 2—“emotional” cluster (“agitation/aggression” and/or “depression/dysphoria” and/or “anxiety” and/or “irritability” items); and 3—“behavioral” cluster (“euphoria/elation” and/or “apathy” and/or “disinhibition” and/or “aberrant motor behavior” items) [13]. The study showed no statistically significant impact of different neuropsychiatric sub-syndromes on the rate of cognitive decline, indicating that the cognitive progression of dementia seems to be scarcely affected by the presence of specific clusters of symptoms [13]. Thompson et al. examined the associations between dementia subtypes, severity of dementia and severity of BPSD. They found that severity of BPSD did not differ between AD and vascular dementia, but was higher in those patients with greater severity of dementia [14]. Considering that different behavioral symptoms belonging to different clusters are often co-presenting, the idea that there could be a common underlining neurotransmitters disruption may arise. Monoamine 5-hydroxytryptamine (5-HT), or serotonin, is one of the most important neurotransmitters in the central nervous system (CNS), regulating multiple physiological functions. 5-HT works as both a neurotransmitter and neuromodulator, acting in both central and peripheral systems. Serotonergic circuitry has been tied to cognitive decline and implicated in a number of basal and higher brain functions that are perturbed in BPSD. It is highly possible that the co-clustering of BPSD into domains depends on different circuits via diverse expression of 5-HT receptor subunits [15]. Dopaminergic system as well is involved in behavioral disturbances genesis and control. Dopamine (DA) is not only fundamental for motor control, due to the activity within the basal ganglia, but is also responsible for the processing of cognitive information, perception and adaptation to the environment [16].

## 3. Antipsychotic Use in Dementia

Antipsychotics represent the main pharmacological strategy to alleviate BPSD, improving the quality of patients’ and caregivers’ lives [17]. Despite the warnings issued by the US Food and Drug Administration (FDA), the European Medicines Agency and the UK Medicines and Healthcare Products Regulatory Agency, antipsychotics are often used in individuals with dementia for sustained periods (≥6 months) [18,19], although they are associated with increased risk of death, cerebrovascular adverse events (CVAEs), Parkinsonism, sedation, gait disturbance, cognitive decline and pneumonia [20]. This risk remains elevated for at least 2 years, with an increased number of deaths due to antipsychotics prescription and longer duration of use. A recent meta-analysis including several large retrospective studies showed an increased all-cause mortality associated with antipsychotic use in patients with dementia [21]. Nevertheless, these drugs have been increasingly prescribed over the last several years, even for long-term use. Amongst antipsychotics, only risperidone is indicated for the short-term management of persisting and severe aggression in individuals with AD who have failed non-pharmacological trials [22]. Off-label treatment with antipsychotic medications has grown in the past two decades, with increasing prescription rates, estimated to be between 20% and 50% [23], and is even higher among institutionalized individuals with dementia [24]. Given that, the importance of the careful evaluation of the potential drug–drug interactions between antipsychotics and Ache-I or memantine, especially in a population of older patients often affected by several chronic diseases and undertaking polytherapy, becomes clear [25]. The complex management of BPSD requires a deep knowledge of antipsychotics’ mechanism of action, possible pharmacological interactions, symptoms overlapping and spectra, without overlooking the social context, the patient and caregiver counselling and always considering a non-pharmacological therapy as a first approach. The most relevant literature for the possible mechanism of action of antipsychotic drugs derives from studies in schizophrenia and mania; even though it is possible to apply them to a geriatric population from a purely pathophysiological point of view, it is mandatory to fully assess the older and often frail demented patient, in order to minimize adverse events or drug–drug interactions.

Antipsychotics are commonly classed as either typical or atypical based upon their potency as dopamine D2 receptor antagonists and their actions on serotonin 5-HT2A receptors [26]. While several studies, such as the National Institute of Mental Health (NIMH), Clinical Antipsychotic Trials of Intervention Effectiveness (CATIE) and sub-studies have not demonstrated a clear and significant difference between second and first generation antipsychotics, at least for schizophrenia, their better safety profile, particularly for extrapyramidal symptoms (EPS), would grant them some actual advantage [27]. 

### 3.1. Typical Antipsychotics: Mechanisms and Limitations 

Based on their chemical structures, they are grouped into several classes: phenothiazines (e.g., chlorpromazine and fluphenazine), butyrophenones (e.g., haloperidol), benzamides (e.g., sulpiride and tiapride), and were developed mainly for the treatment of schizophrenia, with the first agents licensed in the 1950s. Since their discovery, first generation antipsychotics have been the standard for treating psychotic disorders for many decades. These classical neuroleptics or typical antipsychotics display a rather narrow spectrum of therapeutic activity, however, and because of their wide receptor profile, their use is associated with several side effects. Since they bind predominantly to D2 receptors throughout the brain as powerful, long-lasting antagonists, as well as to a broad range of other receptors, including D1, 5-HT2, histamine H1 and α2 adrenergic receptors [26], they lack the tolerability of newer antipsychotics, inducing, among others, sedation, anticholinergic effects and EPS. Haloperidol is now the most widely prescribed agent from this category, mainly because it is considered a first-line treatment in hypoactive, hyperactive and mixed type delirium, according to NICE guidelines [28]. Haloperidol preferentially binds dopamine receptors (in particular, D2, D3 and D4) and α1 adrenergic receptors, while it has negligible affinity for H1, M1 and 5-HT (in particular, 5-HT 2C) receptors [26]. Although a recent study [29] reported that the number needed to harm (NNH) with haloperidol for the outcome of mortality was similar to risperidone, another more recent study evaluating community dwelling AD patients’ mortality documented a higher mortality risk in patients treated with haloperidol compared to quetiapine or risperidone [30]. When a diagnosis of dementia was not required for inclusion, risk of both death and femur fracture in nursing home residents was higher for conventional antipsychotics compared with atypical antipsychotics [31]. Recently, a large retrospective study conducted on a population of patients with newly diagnosed dementia evaluated the impact of antipsychotic medications on acute cerebral and cardiovascular events, hip fracture and venous thromboembolism [32]. The use of antipsychotic drugs appeared to be associated with increased risk of stroke, thromboembolism and hip fracture, while no increased risk was detected regarding long-term mortality [32]. In addition, a more recent systematic review of 36 Randomized Clinical Trials (RCTs) compared the efficacy of risperidone, haloperidol, SSRI as a class and dextromethorphan/quinidine in treating agitation in people affected by all-types dementias; the results showed that haloperidol was almost the least efficacious among all comparators, dissuading prescription of this medication in this particular case [33]. Lastly, the American Psychiatric Association (APA) recommends that in the absence of delirium, if nonemergency antipsychotic medication treatment is required, then haloperidol should not be used as a first-line agent (Recommendation 1B). Furthermore, the APA recommends that in individuals with dementia and agitation or psychosis a long-acting injectable antipsychotic medication should not be utilized unless it is otherwise indicated for a co-occurring chronic psychotic illness (Recommendation 1B) [34]. 

### 3.2. Atypical Antipsychotics: Mechanisms and Advantages/Limitations

Atypical antipsychotics include clozapine, risperidone, olanzapine, quetiapine and aripiprazole. They comprise serotonin and dopamine antagonists (SDAs), multiple-acting receptor targeted antipsychotics (MARTAs) and dopamine D2 partial agonists [35]. 

It must be noted, however, that these second generation antipsychotics (SGA) target a broader range of receptors with different affinity. In general, they not only exert antagonist effect on dopamine D2, but also have a simultaneous antagonist effect on 5-HT receptors, particularly on the 5-HT_2A_; this results in increased blockage efficacy on the mesolimbic pathways, but not on the nigrostriatal one [26].

However, different potency of affinity splits them into a group of drugs with modest affinity for D2, 5-HT_2A_ and other receptors such as H_1_ and M_1_ (clozapine, olanzapine and quetiapine) and those with potent antagonist action on D2 and 5-HT_2A_, high affinity for α_1_, 5-HT_2c_ and H_1_ and negligible affinity for M_1_ receptors (risperidone, paliperidone, lurasidone) [26]. Clozapine has become the prototype for new neuroleptics, due to its favorable receptor profile and low incidence of Parkinsonism and tardive dyskinesia; however, the increased risk for agranulocytosis, weight gain and metabolic alterations had a negative impact on its use. Risperidone, with higher affinity for 5-HT_2A_ than for D2, has shown good efficacy in treating positive symptoms and increased dopaminergic neurotransmission in the nigrostriatal pathway with reduced EPS [36]. However, the strong binding to 5-HT_2C_, α_1_ and H_1_ is responsible for the side effects, such as weight gain, sedation, orthostatic hypotension [26]. The higher affinity for different target receptors justifies the possible different or added desired or adverse effects of the different drugs. In particular, affinity to histamine-1 receptor is higher for olanzapine and quetiapine, to 5HT-5_1C_ for risperidone, clozapine and olanzapine, to adrenergic receptor for clozapine, quetiapine, olanzapine (α1 and α2) and risperidone (α2) [37]. A further improvement in their mechanism of action led to the development of a third generation of antipsychotics. Often referred to as dopamine system stabilizers (DSSs), they act as partial D2, D3 and 5-HT_1A_-receptor agonists, and antagonists at 5-HT_2A_ receptors. In other words, they can act either as a functional agonist or a functional antagonist, depending on the surrounding levels of dopamine. The antipsychotic action would follow the functional antagonism in the mesocortical pathway, where excess of dopamine causes positive symptoms, while the action as functional agonist in the mesocortical pathway improves the negative symptoms [38]. Reliant on local levels of dopamine, DSSs do not cause motor side effects, preserving dopamine activity in those regions where normal dopamine levels are needed (nigrostriatal pathway) [26]. Aripiprazole may be considered representative of this latter group of neuroleptics, with its reduced association with extrapyramidal side effects and its efficacy against both positive and negative symptoms of schizophrenia. Aripiprazole causes minimal weight gain, sedation and does not produce elevation in serum prolactin levels; most importantly, unlike other neuroleptics, it does not lengthen QTc interval on electrocardiogram [39]. Nonetheless, atypical neuroleptics account for >80% of the neuroleptics prescribed for people with dementia, and the most widely prescribed are risperidone, olanzapine and quetiapine. Several studies compared first generation antipsychotics (FGA) and SGA safety profile. In 2014, the increased risk of cardio and cerebral vascular events (stroke, ventricular arrhythmia, myocardial infarction), as well as hip fractures, has been highlighted [40]. Almost 10% of strokes and hip fractures were more frequent in the group treated with FGAs, whereas the difference in the two groups was lower for myocardial infarction and ventricular arrhythmia. Combining these data, all the adverse events accounted for approximately one sixth of the mortality differences between FGAs and SGAs, even though this difference could be as large as 42% [40]. Recently, a systematic review analyzed a total of 16 meta-analyses evaluating the use of antipsychotics in individuals with dementia; of those, only two were specifically focused on AD, one on LBD and the others more generically on dementia. The authors did not find any evidence in the evaluation of the difference in mortality rates between first and second generation antipsychotics (FGAs and SGAs) in older adults [41]. In particular, 10 meta-analyses evaluated atypical antipsychotics and only two meta-analyses evaluated typical antipsychotic medications. When used in individuals with dementia, including AD, atypical antipsychotic medications, especially quetiapine, showed modest efficacy. Greater responses to atypical antipsychotics were observed in individuals with severe psychosis, aggression and agitation, whereas smaller effects were noted for subjects with less severe symptoms. Furthermore, in these 10 meta-analyses, antipsychotics use in individuals with dementia was associated with a greater number of adverse effects when compared with individuals treated with placebo, including the risk of CVAEs and death. 

In comparative effectiveness studies of second generation antipsychotics, risperidone was superior to quetiapine in the Cohen-Mansfield Agitation Inventory (CMAI) [42]. The effectiveness of quetiapine is considerably weaker than risperidone. Nevertheless, a meta-analysis involving five randomized trials observed a statistically significant effect relative to placebo on neuropsychiatric symptoms, as evaluated with the NPI and overall improvement (Clinical Global Impression (CGI) scores) [43]. Moreover, quetiapine could be the antipsychotic of use to treat BPSD in patients with Parkinsonian features, thanks to the lower induction of extrapyramidal signs [8]. Aripiprazole shows a weaker effectiveness than risperidone as well. Moreover, aripiprazole showed better outcome, compared to placebo, in NPI, Brief Psychiatric Rating Scale (BPRS) and Cohen-Mansfield Agitation Inventory (CMAI), while olanzapine, quetiapine and risperidone did not [42]. Even though atypical antipsychotics have a better safety profile, they may present with several adverse events, such as anticholinergic effects, orthostatic hypotension, seizures, metabolic syndrome, weight gain, extrapyramidal symptoms, sedation and QT-prolongation [8]. Another important finding is the increased risk of stroke and mortality associated with the use of atypical antipsychotics [44]. 

Notwithstanding the slight advantage of second upon first generation antipsychotics, in the recent network meta-analysis pooling together studies mainly on dementia, but also including mixed and one study comprising LBD, no atypical antipsychotic was consistently associated with better results than the others across all effectiveness and safety outcomes, risk of death included [42].

All these data suggest the importance of being cautious with the prescription of antipsychotics, particularly in frail patients, where the increased risk of hip fracture or cardiovascular events might accelerate or worsen the loss of independence, increase the hospitalization and the global outcome as well as the cognitive impairment. An accurate medical history and global comprehensive medical assessment would reduce the inappropriate prescription of drugs such as olanzapine or risperidone in patients with high cardiovascular risk or cerebral ischemia, as suggested also by the American Psychiatric Association guidelines and STOPP/START criteria.

## 4. Antipsychotic Therapy: Management Issues and Drugs Switch 

The first issue for geriatricians having to treat BPSD in patients with dementia is the non-optimal risk/benefit balance of antipsychotics use in these patients. Unfortunately, however, the incidence of psychiatric symptoms is high and the need for treatment to manage patients and caregivers, avoiding inappropriate hospital admission and further functional and cognitive decline, becomes clear. It must be highlighted that only two drugs are licensed for BPSD treatment in dementia: pimavanserin for hallucinations and delusions associated with Parkinson’s Disease (PD) psychosis (USA), and risperidone for short-term treatment of persistent aggression in moderate-to-severe AD (only in Canada and UK). In the remaining cases, the use of antipsychotics is off-label and guided by the physician’s judgment [45].

A glance at the real word shows that up to 60% of patients with cognitive impairment in hospitals and long-term care homes are being treated with antipsychotics for neuropsychiatric symptoms related to dementia and AD [46]. The majority of the antipsychotics administered are the atypical ones. These drugs, especially risperidone, aripiprazole and olanzapine, have been evaluated in multiple studies and show improvement in symptoms like severe agitation, aggression and psychosis (such as delusions and hallucinations) in patients with dementia [41]. Risperidone and aripiprazole are the most employed drugs in BPSD and they are effective in the treatment of psychotic symptoms, agitation and aggression [47,48]. 

The incidence of side effects observed in previous studies, the poor tolerability and the increased risk of mortality related to the antipsychotic therapy demand further research [46]. The second issue for clinicians is the choice of the right drug and therapy-tailoring; as a general guideline, it is desirable to use antipsychotics in the lowest dose sufficient to control symptoms, for the shortest duration, with close monitoring for the development of adverse effects [49]. Moreover, regular clinical evaluations of risks and benefits are necessary during the treatment [34]. To decrease the risks mentioned above, physicians should consider gradual tapering off of antipsychotics once control of behavioral symptoms is achieved, although there is limited evidence regarding patient outcomes after stopping antipsychotics [49] and insufficient evidence to indicate whether discontinuation impacts on mortality or other side effects associated with antipsychotics [50]. As underlined in a recent Cochrane, the evidence that the interruption of long-term antipsychotic therapy in older patients with dementia may be done without worsening of symptoms is limited; in particular, higher benefit may be seen in patients with milder symptoms [50]. 

To facilitate the process of tapering off the medication, Tjia et al. proposed a gradual reduction of the dose of the antipsychotic in two steps based on pharmacokinetic principles that favor drug discontinuation [51]. A small study of 36 patients suggested that the use of citalopram may facilitate the withdrawal of antipsychotics for the elderly with AD [52]. The Halting Antipsychotic Use in Long-term Care (HALT) study was a single-arm longitudinal study conducted in Australian long-term care facilities among patients taking antipsychotics, 98.5% of whom had dementia. Of the 93 patients who completed the study, 69 (74%) had antipsychotics successfully deprescribed without reinitiating antipsychotics or experiencing increase in BPSD [53]. A question of primary importance, relating to the use of these molecules, is the management of the switch from one antipsychotic to another. There are essentially two reasons that can lead a clinician to program a switch: suboptimal efficacy of a drug, resulting in the need to switch to a different drug, or appearance of unacceptable side effects for the doctor and/or the patient, associated with a reduction in adherence to therapy and a risk of potential relapse. The transition between different antipsychotics can often be associated with the onset, albeit transient, of unwanted clinical manifestations. In particular, if the switch is not done correctly, unwanted effects due to the rebound effect and withdrawal symptoms can occur [54]. The risk of rebound side effects is particularly high following abrupt discontinuation of a short half-life antipsychotic and replacing it with a longer half-life antipsychotic. The likelihood of rebound and withdrawal effects is greater when the two antipsychotics (pre- and post-switch) differ from each other in their pharmacodynamic profile (pharmacodynamic rebound) or when the half-life of the pre-switch antipsychotic is particularly short (pharmacokinetics rebound).

Pharmacokinetic rebound occurs when the post-switch antipsychotic is relatively underdosed, that is, when its plasma levels are not sufficiently high to achieve a degree of functional receptor blockage similar to that obtained by the pre-switch drug. In clinical terms, the outcome of pharmacokinetic rebound is identical to that of pharmacodynamic rebound. In general, the lower the half-life of an antipsychotic that must be discontinued and replaced with another antipsychotic, the greater the risk of pharmacokinetic rebound effects upon its abrupt withdrawal, especially if the second drug has a longer half-life [55]. A pharmacodynamic rebound occurs when patient’s receptors, previously exposed for a prolonged period to a blocking action by a specific antipsychotic, with consequent effect of upregulation, suddenly find themselves exposed to the endogenous ligand for that type of receptor. This can occur when an antipsychotic is interrupted or abruptly switched to another one with less affinity towards that same receptor system. The rebound effects experienced by the patient are generally opposite to those attributable to the receptor block. 

The rebound following the suspension of dopaminergic drugs is due to hypersensitivity to endogenous dopamine, and the patient might present either with psychosis or supersensitivity mania due to the effect on the mesolimbic system, or with rebound dyskinesia due to the effect on the nigrostriatal system [56]. The histaminergic rebound effect usually follows the abrupt withdrawal of an antipsychotic with a potent histaminergic blocking action, as in the case of chlorpromazine, clozapine, quetiapine and olanzapine. Since the blockage of the H1 is associated with anxiolytic, sedative effect, sleep induction, increased appetite and weight, an abrupt discontinuation of one of the previously mentioned antipsychotics could lead to the onset of agitation, anxiety and insomnia [54]. Finally, the cholinergic rebound occurs in cases of discontinuation of an antipsychotic with cholinergic blocking action, such as clozapine, olanzapine and quetiapine. The abrupt transition, followed by excessive stimulation of the central M1 receptors, can trigger rebound symptoms such as agitation, insomnia, mental confusion, psychosis, anxiety, drooling, EPS/akathisia together with diarrhea, sweating, nausea, vomiting [55]. Table 1 shows the possible rebound symptoms when interrupting a drug, depending on specific receptors’ blockage release. 

To prevent or at least reduce the intensity of withdrawal and rebound symptoms during the transition to antipsychotic treatment, several strategies are available. These include the use of benzodiazepines, antihistamines, anticonvulsants, anticholinergics, beta blockers or the prolongation of the switching phase and the application of the targeted switching strategy based on the specificities of the molecules involved in the switch [57]. Unfortunately, even in the case of switching strategies, the literature is scarce and usually about schizophrenia or schizoaffective disorders [58], while dementia is under-represented, especially regarding recent literature. A systematic review and meta-analysis evaluated the available literature regarding immediate versus gradual antipsychotic switch in patients with schizophrenia [59]. In this meta-analysis, no difference was found in terms of several clinical outcomes between the two groups (immediate vs. gradual discontinuation when switching drug). That finding is partially at odds with the previous literature, suggesting gradual discontinuation in order to mitigate rebound effect; however, the lack of difference could be due to the different pharmacological properties other than towards dopamine receptors, as well as to the dose equivalents of the new drugs [59]. A review published by Cerovecki et al. evaluated the antipsychotic switching, but again on a population with schizoaffective disorders [54]. In this review, few examples of drug switching have been reported, with relevant diverse events. However, the switches reported should be read with a critical approach by the clinician treating BPSD in dementia, since the populations examined are hugely different, some of the studies were sponsored, and few of the reported papers were case reports of purely psychiatric subjects [54]. The finding from the broad literature review, which could certainly be useful, was that exacerbation of psychosis was more frequent when switching from clozapine or olanzapine to other SGA; rebound psychosis was more evident when switching from the same above mentioned drug to risperidone or aripiprazole [54]. This definitely supports the slow cross-tapering switch, possibly with the support of benzodiazepine or anticholinergic compounds [54]. 

The setting of a specific switching strategy between antipsychotics can greatly contribute in reducing the risk of the phenomenon described above [54,57]. “Abrupt switch” is the simplest strategy in which one drug is immediately replaced with another, with a full and immediate discontinuation of the first antipsychotic drug and subsequent full dose introduction of the second one. This type of switching often occurs in clinical practice, as patients abruptly and autonomously discontinue antipsychotic treatment and then the physician is forced to revert to adequate antipsychotic treatment just as quickly, when he notices the interruption. If, instead, it is planned by the physician, it should be reserved exclusively for cases in which a patient reports a serious adverse event attributable to ongoing antipsychotic therapy, since this strategy is associated with a greater risk of rebound phenomena [55]. “Taper switch” consists of the gradual suspension of the first drug with the immediate start of the second antipsychotic at therapeutic dosage; this modality can be used in cases where the new drug does not require titration. “Cross-taper switch” consists of the gradual suspension of the first drug associated with the progressive increase in the dosage of the new antipsychotic. “Plateau cross-taper switch” consists of the gradual initiation of the new antipsychotic until reaching the full dose, followed by a subsequent gradual withdrawal of the first drug. The period of co-administration of the two molecules must take place for a period of time that allows the new antipsychotic to reach the maximum concentration peak, and therefore be able, when the suspension of the first molecule begins, to bind to the receptor systems [58]. Figure 1 shows the possible drugs-switching schemes. 

Successful antipsychotic switches require planning, time, accurate dosing and knowledge of the pharmacokinetic and pharmacodynamic characteristics of the antipsychotics involved in the switch. Considering the burden of BPSD for patients, caregivers and the healthcare systems, the choice of the best strategy for the drug switching or tapering, whenever necessary, represents a crucial step in the management of a patient with dementia. Strategies requiring a gradual tapering off of the first antipsychotic with a progressive increase of the second drug would probably be the safest in a frail patient, who may have polypathology and be already prescribed with several drugs. Figure 2 shows a possible flow chart when approaching a patient with BPSD, such as agitation, aggression, hallucination or delusions. 

## 5. Conclusions

In recent years, conventional antipsychotics, such as haloperidol, have been replaced by atypical antipsychotics for the treatment of psychosis in dementia. However, they have minimal efficacy on psychotic symptoms and are associated with several adverse effects, such as increased risk of falls and stroke. Nevertheless, the major concern is their association with an increased risk of death. Regulatory agencies warn that these medications should be avoided in older adults with dementia, while practice guidelines from specialist societies recommend that they should be reserved as a second-line treatment for behavioral symptoms, when these patients are unresponsive to non-pharmacological strategies. Among atypical antipsychotics, the use of risperidone, olanzapine and aripiprazole appears to be supported by stronger data when compared with quetiapine for treating BPSD. Even though these medications have reached a nearly acceptable trade-off between effectiveness and safety, a single most effective and safe treatment is yet to be found. The management of psychiatric symptoms in dementia represents an important challenge for clinicians, who have to tailor the therapy according to the patients’ symptoms, comorbidity and polytherapy, bearing in mind the high risk of adverse events; it is recommended to use the lowest dose possible and to attempt therapy discontinuation whenever possible. A standardized treatment algorithm would be desirable, especially considering the prevalence of such a devastating disease.

## Figures and Tables

**Figure 1 pharmaceuticals-14-00246-f001:**
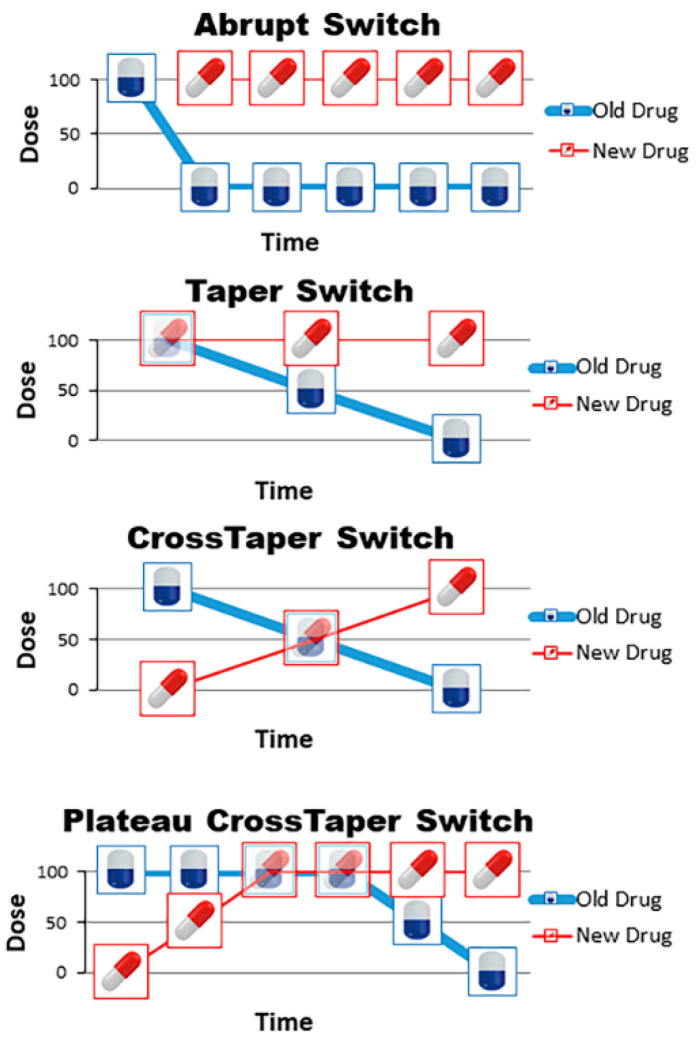
Potential drug-switching schemes. In each panel the old drug is blue, the new drug is red. The x axis represents the timeline and the y axis the percentage of the target dose. The abrupt switch consists of the immediate interruption of the old drug, starting the new drug at full dose. In the taper switch, the old drug is gradually reduced while the new drug is started at full dose. Cross-taper switch consists of gradual reduction of the old drug with gradual increase of the new drug, without overlapping the full doses. Plateau cross-taper switch consists of gradual increase of the new drug, a short overlapping of the two drugs at full dose and then gradual reduction of the old drug.

**Figure 2 pharmaceuticals-14-00246-f002:**
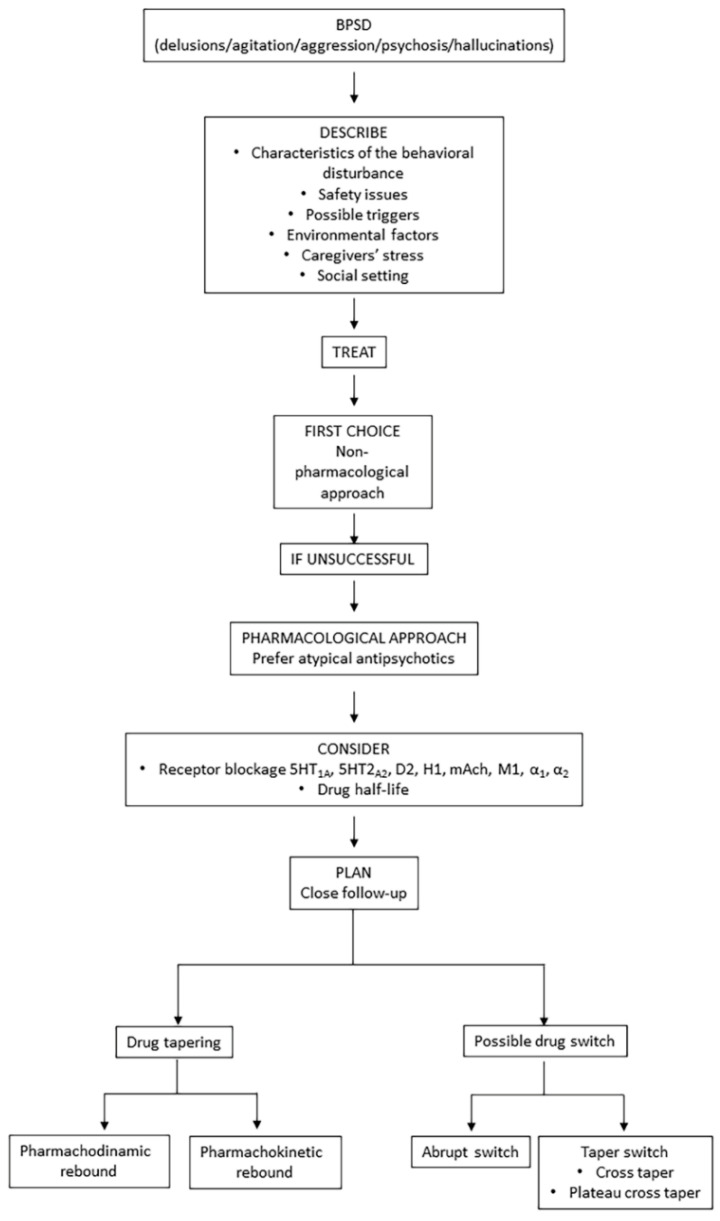
Flow chart for approaching patients with BPSD (in particular delusions/agitation/aggression/psychosis, hallucinations).

**Table 1 pharmaceuticals-14-00246-t001:** Possible rebound symptoms when interrupting an antipsychotic drug.

Receptor	Possible Rebound/Withdrawal Effects
D2	Psychosis, mania, agitation, akathisia, withdrawal dyskinesia
a1 -adrenergic	Tachycardia, hypertension
a2 -adrenergic	Hypotension
H1	Anxiety, agitation, insomnia, restlessness, EPS/akathisia
M1 (central)	Agitation, confusion, psychosis, anxiety, insomnia, sialorrhea, EPS/akathisia
M2-4 (peripheral)	Diarrhea, sweating, nausea, vomiting, bradycardia, hypotension, syncope
5-HT1A	Anxiety, EPS/akathisia
5-HT2A	EPS/akathisia, psychosis
5-HT2C	Anorexia

## Data Availability

Data sharing not applicable.

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
