# Peer review of "Behavioral and Psychological Symptoms in Dementia (BPSD) and the Use of Antipsychotics"

_pharmaceuticals, 2021, doi:10.3390/ph14030246_

Round 1

Reviewer 1 Report

pharmaceuticals-1103991, Behavioural and Psycholgical Symptoms in Dementia (BPSD) and the use of antipsychotics

The paper really needs some figures and tables to add more context and to improve the understanding of the information by the readers. I advise the authors to add some relevant figures and tables to support the review.

The paper also needs a more critical approach. It should be more than a simple collection of data from various sources, but a critical analyze of them. The affiliation of the authors indicates they have clinical experience in this field. They should provide their expert opinion on this subject.

On page 2, it should be risperidone, and not “riperidone”

The authors should verify and better check for each drug mentioned the exact mechanism of action. They presented D2 as target, and “dopamine D2 receptor antagonists “, but the majority of them are antagonist on others dopamine receptors as well. In the same way “blocking action for 5-HT2 receptors” is a clear oversimplification of the actual mechanism. There are many differences between several atypical antipsychotics and it can mislead the readers. In my opinion the authors should highlight in their paper the complex mechanism of action and that there are many differences between apparently similar compounds.

There are long sections with just one reference. The authors should make sure that there is no plagiarism risk. Some references are over 10 years old and could be updated.

Author Response

Comment #1:

The paper really needs some figures and tables to add more context and to improve the understanding of the information by the readers. I advise the authors to add some relevant figures and tables to support the review.

Response #1

We thank the Reviewer for the suggestion and we agree that figures or tables would make the review easier to follow. We have added a table about possible rebound effect (Table 1, page 8 of the revised manuscript) and a figure representing potential drug-switching strategies (Fig 1, page 10 of the revised manuscript).

Comment #2

The paper also needs a more critical approach. It should be more than a simple collection of data from various sources, but a critical analyze of them. The affiliation of the authors indicates they have clinical experience in this field. They should provide their expert opinion on this subject.

Response #2

We thank the Reviewer for the comment. We have now added more personal considerations throughout the manuscript (i.e. page 3 lines 115-117, page 3-4 lines 151-159 and 162-167, page 5 line 243, pages 6 lines 291-298, 300-304, page 7 lines 310, 320-323, page 10 lines 450-452 of the revised manuscript).  All the changes are highlighted in yellow.

Comment #3

On page 2, it should be risperidone, and not “riperidone”

Response #3

We thank the Reviewer for noticing it and we apologize for the typo

Comment #4

The authors should verify and better check for each drug mentioned the exact mechanism of action. They presented D2 as target, and “dopamine D2 receptor antagonists”, but the majority of them are antagonist on others dopamine receptors as well. In the same way “blocking action for 5-HT2 receptors” is a clear oversimplification of the actual mechanism. There are many differences between several atypical antipsychotics and it can mislead the readers. In my opinion the authors should highlight in their paper the complex mechanism of action and that there are many differences between apparently similar compounds.

Response #4

We thank the Reviewer for the Comment. We agree that the mechanisms of action are complex and should not be excessively simplified. However, we did not mean to oversimplify the mechanisms of action; as for the sentence “dopamine D2 receptor antagonists”, we have specified, few lines below, that they bind also “to a broad range of other receptors, including D1, 5-HT2, histamine H1 and α2 adrenergic receptors”, to explain the reduced tolerability. Nonetheless, as requested, we have added more details on the topic (please, see page 4 lines 183-185 and page 5 lines 213-232 of the revised manuscript)

Comment #5

There are long sections with just one reference. The authors should make sure that there is no plagiarism risk. Some references are over 10 years old and could be updated.

Response #5

We thank the Reviewer for the suggestion and, accordingly, we checked the long sentences without references. We have also updated the Bibliography and now most of the references are very recent.

Reviewer 2 Report

By the current population aging rate, number of patients diagnosed with dementia is rapidly increasing and having a comprehensive and accurate knowledge on relevant psychological  symptoms and pharmacological mechanisms/applications of antipsychotics is absolutely necessary. Calsolaro et al. in a their review manuscript tried to cover this important topic. 

I noticed the following points in the submitted manuscript:

1- The manuscript was full of spelling or typing errors which needs to be addressed. For example:

  • In the Abstract, please correct evauated and politherapies to evaluated and polytherapies, respectively.
  • - Page 2, please correct isquite and patiennts to is quite and patients, respectively.
  • -Page 3, correct syntoms to symptoms.
  • -Page 4, correctantipsyhhotic drugs to antipsychotic drugs.
  • -Page 6: correct averse to adverse please.
  • - Page 7: please correct toether to together.

2- In the page 2, the authors mentioned about their pubmed search strategies and did not  point out about the search dates limitations, and it was not known to me why they limited their search to only English language published journals, whereas they could have checked the French, German, Chinese journals (at least their English abstracts) as well.

3- This review needs a Table or Chart to to explain its contents in a better way, although the authors mentioned "Fig 1" on Page 8, but I couldn't find any figure in the manuscript!.

Author Response

By the current population aging rate, number of patients diagnosed with dementia is rapidly increasing and having a comprehensive and accurate knowledge on relevant psychological symptoms and pharmacological mechanisms/applications of antipsychotics is absolutely necessary. Calsolaro et al. in a their review manuscript tried to cover this important topic. 

I noticed the following points in the submitted manuscript:

Comment #1:

1- The manuscript was full of spelling or typing errors which needs to be addressed. For example:

  • In the Abstract, please correct evauated and politherapies to evaluated and polytherapies, respectively.
  • - Page 2, please correct isquite and patiennts to is quite and patients, respectively.
  • -Page 3, correct syntoms to symptoms.
  • -Page 4, correctantipsyhhotic drugs to antipsychotic drugs.
  • -Page 6: correct averse to adverse please.
  • - Page 7: please correct toether to together.

Response #1: we apologize for the several typos. We have gone through the whole manuscript again and corrected accordingly.

Comment #2- In the page 2, the authors mentioned about their Pubmed search strategies and did not  point out about the search dates limitations, and it was not known to me why they limited their search to only English language published journals, whereas they could have checked the French, German, Chinese journals (at least their English abstracts) as well.

Response #2: we thank the Reviewer for the suggestion. We preferred to give priority to full text English papers, but we have now expanded the search with English abstract of French, German and Chinese journals.  We have also added the search dates limits (10 years). Please, see page 2, lines 59-65 of the revised manuscript

Comment #3:- This review needs a Table or Chart to explain its contents in a better way, although the authors mentioned "Fig 1" on Page 8, but I couldn't find any figure in the manuscript!

Response #3: we thank the Reviewer for the suggestion and we apologize for the Figure 1, we had uploaded it in the website, but maybe some error occurred. We have now added also a table and a further figure to the review.

Reviewer 3 Report

This review focuses on the advantages and limitations of antipsychotic drugs usage for treating BPSD in dementia. Undoubtedly, management of BPSD is a challenged due to adverse effects of antipsychotic drugs, limited efficacy some of them and problems in switching between different drugs. Thus, the topic raised by authors is timely and important. However, I have some serious remarks regarding present manuscript, as follows:

Major:

1) Authors should clearly indicate the novelty of their review. It should be noted that the topic of present manuscript is relatively well characterized in some recent review articles (Keks et al., 2019; Bessey and Walaszek, 2019; Magierski et al., 2020; Chen et al., 2021). Moreover, the usage of antipsychotics in treatment of BPSD in Alzheimer’s disease dementia was well covered by authors in their own recent review article (Calsolaro et al., 2019 in Front Pharmacol). The problem of switching between antipsychotic drugs is of great interest as it can help to improve existing treatment strategies. This problem requires a more detailed discussion in the manuscript including data from clinical trials etc. This may help to increase the significance of manuscript.

2) In the introduction authors declare that “In this review we evaluate the use of antipsychotic drugs in Alzheimer’s disease”. At the same time title of the manuscript assumes focus on dementia per se. In addition, authors sometimes discussed findings that not related only to Alzheimer’s disease dementia. As example, discussion of systematic review of Tampi et al. (2016) where only 3 of 16 meta-analyses were conducted in individuals with Alzheimer’s disease was not limited to findings from use of antipsychotics only in Alzheimer's disease. So, authors should either limit discussion to addressing Alzheimer's disease dementia or expand the review to include any available information on the use of antipsychotics in the treatment of BPSD in other types of dementia.

Minor:

Document should be edited thoroughly for spelling and grammar. The manuscript contains a lot of misprints (starting from title).

Author Response

Major:

Comment #1 Authors should clearly indicate the novelty of their review. It should be noted that the topic of present manuscript is relatively well characterized in some recent review articles (Keks et al., 2019; Bessey and Walaszek, 2019; Magierski et al., 2020; Chen et al., 2021). Moreover, the usage of antipsychotics in treatment of BPSD in Alzheimer’s disease dementia was well covered by authors in their own recent review article (Calsolaro et al., 2019 in Front Pharmacol). The problem of switching between antipsychotic drugs is of great interest as it can help to improve existing treatment strategies. This problem requires a more detailed discussion in the manuscript including data from clinical trials etc. This may help to increase the significance of manuscript.

Response #1: We thank the Reviewer for requiring more detailed discussion about the therapy switching, which certainly would be the novelty of the review. We have reduced the length of the previous studies description, since it has been already well covered in other recent reviews, and we have implemented the geriatrician/clinician point of view when facing these drugs. Moreover, we have expanded the section about therapy switching; unfortunately, the literature is very scanty for patients with dementia, and most of the studies on that topic come from psychiatric settings, such as schizophrenia, mania and schizoaffective disorders. We have described what could be borrowed from psychiatric disorders and adapted to geriatric patients, stressing the difference between two quite different clinical entities (please, see page 9 lines 396-417 of the revised manuscript). The changes we made are highlighted in yellow.

Comment #2: In the introduction authors declare that “In this review we evaluate the use of antipsychotic drugs in Alzheimer’s disease”. At the same time title of the manuscript assumes focus on dementia per se. In addition, authors sometimes discussed findings that not related only to Alzheimer’s disease dementia. As example, discussion of systematic review of Tampi et al. (2016) where only 3 of 16 meta-analyses were conducted in individuals with Alzheimer’s disease was not limited to findings from use of antipsychotics only in Alzheimer's disease. So, authors should either limit discussion to addressing Alzheimer's disease dementia or expand the review to include any available information on the use of antipsychotics in the treatment of BPSD in other types of dementia.

Response #2: we thank the Reviewer for their comments and suggestions for the improvement of the manuscript. We acknowledge that the data presented are mostly about AD, as we actually specify in the Introduction, while the review covers dementia per se. We have therefore amended the introduction, leaving “dementia” rather than Alzheimer’s disease. However, the most recent literature about the use of antipsychotics in BPSD is scarce and most of the studies are conducted on patients with AD; the meta-analysis and systematic review cited pooled together several different studies where only few of them were, for example, LBD. For this reason, we decided to keep “dementia” per se throughout the whole review, but we have updated the relevant sections specifying the dementia type, if and when available. Please, see pages 6 lines 259-262, 271-285, 287-288 of the revised manuscript).

Round 2

Reviewer 1 Report

The authors made significant modifications according to the comments of the reviewers and improved the quality of the paper. It can be published after some editorial check.

Author Response

We thank the Reviewer for the feedback. We will go through all the required further checks

Reviewer 2 Report

The revised version of the review manuscript improved significantly. It looks good to me except the Figure 1, needs a short descriptive legend to be added to the title of the figure. Thank you.

Author Response

We thank the Reviewer for the positive feedback and for the further suggestion. We have now added a legend to make Fig 1 clearer (Page 12, highlighted in yellow)

Reviewer 3 Report

The authors have adequately addressed concerns that this reviewer raised previously. No further concerns 

Author Response

We thank the Reviewer for the positive feedback about the Manuscript.